# Facial Artery Myomucosal Flap vs. Islanded Facial Artery Myomucosal Flap Viability: A Systematic Review

Giorgio Lo Giudice [1], Romolo Fragola [2], Giovanni Francesco Nicoletti [3], Gabriele Cervino [4,*], Eugenio Pedullà [5], Nicola Zerbinati [6] and Raffaele Rauso [2]

1. Department of Neurosciences, Reproductive and Odontostomatological Sciences, Maxillofacial Surgery Unit, University of Naples "Federico II", Via Pansini, 5, 80131 Naples, Italy; giorgio.logiudice@gmail.com
2. Multidisciplinary Department of Medical-Surgical and Dental Specialties, Oral and Maxillofacial Surgery Unit, University of Campania "Luigi Vanvitelli", Via Luigi De Crecchio, 6, 80138 Naples, Italy; romolofragola@gmail.com (R.F.); raffaele.rauso@unicampania.it (R.R.)
3. Multidisciplinary Department of Medical-Surgical and Dental Specialties, Plastic Surgery Unit, University of Campania "Luigi Vanvitelli", Via Luigi De Crecchio, 6, 80138 Naples, Italy; giovannifrancesco.nicoletti@unicampania.it
4. Department of Biomedical and Dental Sciences, Morphological and Functional Images, University of Messina, 98100 Messina, Italy
5. Department of General Surgery and Surgical-Medical Specialties, University of Catania, 95124 Catania, Italy; eugeniopedulla@gmail.com
6. Dermatology Department, University of Insubria, Via Guicciardini, 9, 21100 Varese, Italy; nzerbinati@centro-medico.it
* Correspondence: gcervino@unime.it

**Abstract:** The aim of this study was to estimate the overall viability of the islanded facial artery myomucosal flap (iFAMM) compared to the facial artery myomucosal flap (FAMM). A systematic review of English articles was performed on PubMed and Cochrane Library electronic databases. Search terms included every nomenclature variant for FAMM flap and iFAMM flap. A total of 373 articles were identified, and 20 articles were considered eligible for inclusion in the review. A total of 486 flaps were evaluated (350 FAMM and 136 i-FAMM flaps). In all studies, the primary outcome assessed was flap viability, collecting the rate of total and partial flap necrosis and then comparing FAMM to i-FAMM. Overall reported partial/total necrosis rate for FAMM flap was 9.7%, 1.4% as total and 8.3% as partial necrosis. Overall partial/total reported necrosis rate for iFAMM flaps was 2.2%, 1.5% as total and 0.7% as partial necrosis. FAMM flaps, both as classical or islanded variants, are an effective option for intraoral small/medium sized defect reconstruction. Outcomes from the present review show a higher partial/total survival rate when this flap is harvested as islanded flap.

**Keywords:** FAMM; i-FAMM; viability; facial artery musculomucosal flap; oral oncology; oral reconstruction

## 1. Introduction

### 1.1. Rationale

Intraoral mucosal defects can be related to several pathological states such as cancer ablation, palatal cleft and oronasal fistula following conventional closure. Several surgical techniques, involving local, loco-regional and free flap have been described in literature in order to reconstruct intraoral defects in areas such as the tongue, the soft and hard palate, the floor of the mouth and lips [1–3]. Microsurgical tissue transfer is often identified as the gold standard in surgical reconstruction, although it requires prolonged operating time, longer hospitalization and, especially for intraoral defects, makes "like-with-like" reconstruction difficult. In cases of small or medium sized intraoral defects, the use of cheek mucosa flap was shown to be effective. Bozola et al. was the first to describe in 1989

an axial buccal mucosal flap based on the buccal artery for the resurface of oral mucosa defects [4]. In 1992, Pribaz combined the principles of nasolabial and buccal mucosa flaps in order to describe an axial musculomucosal flap designed on the facial artery: the facial artery myomucosal flap, also known as FAMM flap [5]. Although similar in concept to Bozola's axial flap, the course of the facial artery, and therefore the orientation of the entire FAMM flap, has several advantages over buccal artery based flaps, allowing the FAMM flap to be more versatile in closing both upper (palate and lateral oropharinx wall) and lower (floor of the mouth, tongue, etc.) intraoral defects. Moreover, the direct and indirect artery flow variants are additional surgical choices that increment the flap versatility. One of the biggest disadvantages of the FAMM flap is the need for tooth extraction or bite block in order to avoid pedicle bite before the second surgical step, usually performed 3 weeks after the preceding surgery, represented by the section of the pedicle and flap remodeling.

In 1999, Zhao et al. described the "island" variant of the FAMM, the so-called i-FAMM flap [6]. This flap has the advantage to avoid the need for temporary bite block after the operation and the need for late pedicle division.

## 1.2. Objectives

In the present systematic literature review, we evaluated flap viability comparing FAMM to i-FAMM flap in oral reconstruction.

## 2. Materials and Methods

### 2.1. Protocol and Registration

The methods and the inclusion criteria of this work were specified in advance and documented in a protocol, according to quality standards described in the PRISMA 2009 checklist [7]. The review was registered in the CRD York website PROSPERO (protocol number CRD42020210349).

### 2.2. Eligibility Criteria

The following focus question was developed according to the population, intervention, comparison and outcome (PICO) study design: In patients undergoing oral reconstruction surgery (P), does i-FAMM flap (I) compared to FAMM flap (C) have better viability (O)?

### 2.3. Information Sources

The research was carried out on electronic PubMed and Cochrane Library databases identifying articles from September 1992 to September 2020. The search was conducted up to September 4th, 2020. Article language was limited to English using database-supplied filters.

### 2.4. Search

The keywords were used and combined with Boolean operators, adapted for every database, both as text words and Medical Search Headings (MeSH terms) as follows: FAMM flap; facial artery myomucosal flap; facial artery musculomucosal flap; i-FAMM; islanded facial artery myomucosal flap; islanded facial artery musculomucosal flap; Bozola flap; myomucosal cheek flap; musculomucosal cheek flap; Zhao flap [title] AND facial artery; buccinator myomucosal flap; buccinator musculomucosal flap; t-FAMMif; tunnelized facial artery myomucosal flap; tunnelized facial artery musculomucosal flap.

### 2.5. Study Selection

The full texts of all possibly relevant studies were selected considering the following inclusion criteria: techniques studies; retrospective studies; prospective studies; articles written in English. Exclusion criteria were: articles where outcomes related to flap viability were not reported; articles where numbers of patients were not described; articles based on buccal flap harvested with no pedicle identification; cadaveric or animal studies. Case reports and case series with fewer than 10 patients were excluded due to the insufficient information provided by a limited number of patients. Review articles were excluded, but

their reference lists were examined to identify other potentially pertinent studies; editorials, letters and commentaries were excluded. Two reviewers (R.F., G.L.G.) performed eligibility assessment independently. Disagreements between reviewers were resolved by consensus. When consensus was not reached, a senior member mediated (R.R.).

### 2.6. Data Collection Process

Two reviewers (R.F., G.L.G.) performed data extraction independently. Disagreements between reviewers were resolved by consensus. When consensus was not reached, a senior member mediated (R.R.). A standard chart form of the obtained data was prepared to facilitate comparison among the articles.

### 2.7. Data Items

The following data from each study were extracted: author, date, study design, number of patients, number of iFAMM and FAMM flaps performed, rate of partial and total flap necrosis.

### 2.8. Risk of Bias in Individual Studies

Two independent reviewers (G.L.G., R.F.) performed quality assessments of the included studies; in cases of discrepancies in the results, they consulted a third senior reviewer (R.R.). ROBINS-I tool was be used to assess non-randomized studies. Five levels (low, moderate, serious, critical or no information) were used to present the risk of bias [8]. The Robvis visualization tool web app was used to create "traffic light" plots of the domain-level judgements for each individual result and weighted bar plots of the distribution of risk-of-bias judgements within each bias [9].

### 2.9. Summary Measures

Partial and total flap necrosis were expressed as integer numbers and percentage.

### 2.10. Additional Analyses

No additional analyses were performed.

## 3. Results

### 3.1. Study Selection

The PubMed and Cochrane Library database search identified 373 articles: 156 were duplicate articles, while 217 studies were screened for title, abstract and language. Seventy-two full-text articles were finally selected for further evaluation. Of the 72 papers, 49 were excluded: 4 case reports, 13 case series with fewer than 10 patients, 3 letters, 1 review article, 1 paper in Spanish language, 21 papers reporting other kinds of flap than FAMM or i-FAMM, 5 papers not reporting flap viability outcomes and 4 papers not meeting the inclusion criteria. The selection process identified 20 articles as eligible for inclusion in the review: 12 retrospective studies, 5 case series and 3 case-control studies (Figure 1).

### 3.2. Study Characteristics

The review process identified a total of 20 studies: Pribaz et al. (1992); Zhao Z et al. (1999); Ashtiani et al. (2005); Joshi A. et al. (2005); Lahiri A. et al. (2007); Ayad et al. (2008); Bianchi B et al. (2009); Massarelli et al. (2012); Shetty et al. (2013); Ferrari et al. (2015); Ferrari et al. (2015); Lee et al. (2016); Sohail et al. (2016); Ahn et al. (2017); Massarelli et al. (2017); Ibrahim B. et al. (2018); Asairinachan et al. (2019); Janardhan et al. (2020); Benjamin et al. (2020); Joseph et al. (2020) [5,6,10–27].

Participants

A total of 486 flaps were evaluated, 350 FAMM and 136 i-FAMM flaps. In all studies, primary outcome assessed was flap viability, collecting the rate of total and partial flap necrosis and then comparing FAMM to i-FAMM.

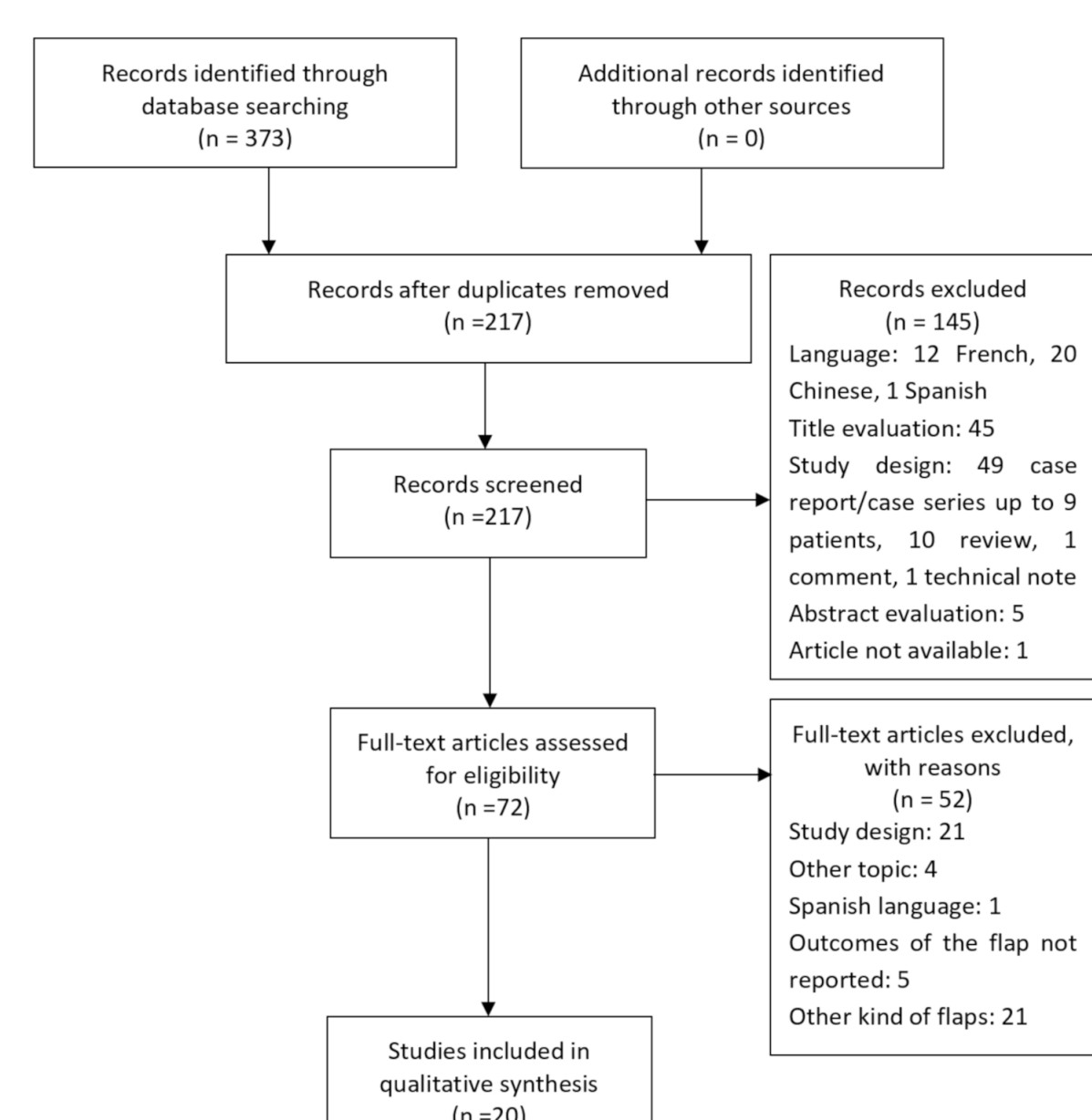

**Figure 1.** Flow diagram of literature search and study selection.

### 3.3. Risk of Bias within Studies

The analysis of the paper quality assessment is presented in Figure 2; Figure 3.

### 3.4. Synthesis of Results

Data extraction from the 20 articles evaluated allowed us to list a total of 350 FAMM and 136 i-FAMM flaps performed. Overall reported necrosis rate for FAMM flaps was 9.7%, 1.4% (5 cases) as total and 8.3% (29 cases) as partial necrosis. Overall reported necrosis rate for iFAMM flaps was 2.2%, 1.5% (2 cases) as total and 0.7% (1 case) as partial necrosis. Results are synthesized in Table 1; Table 2.

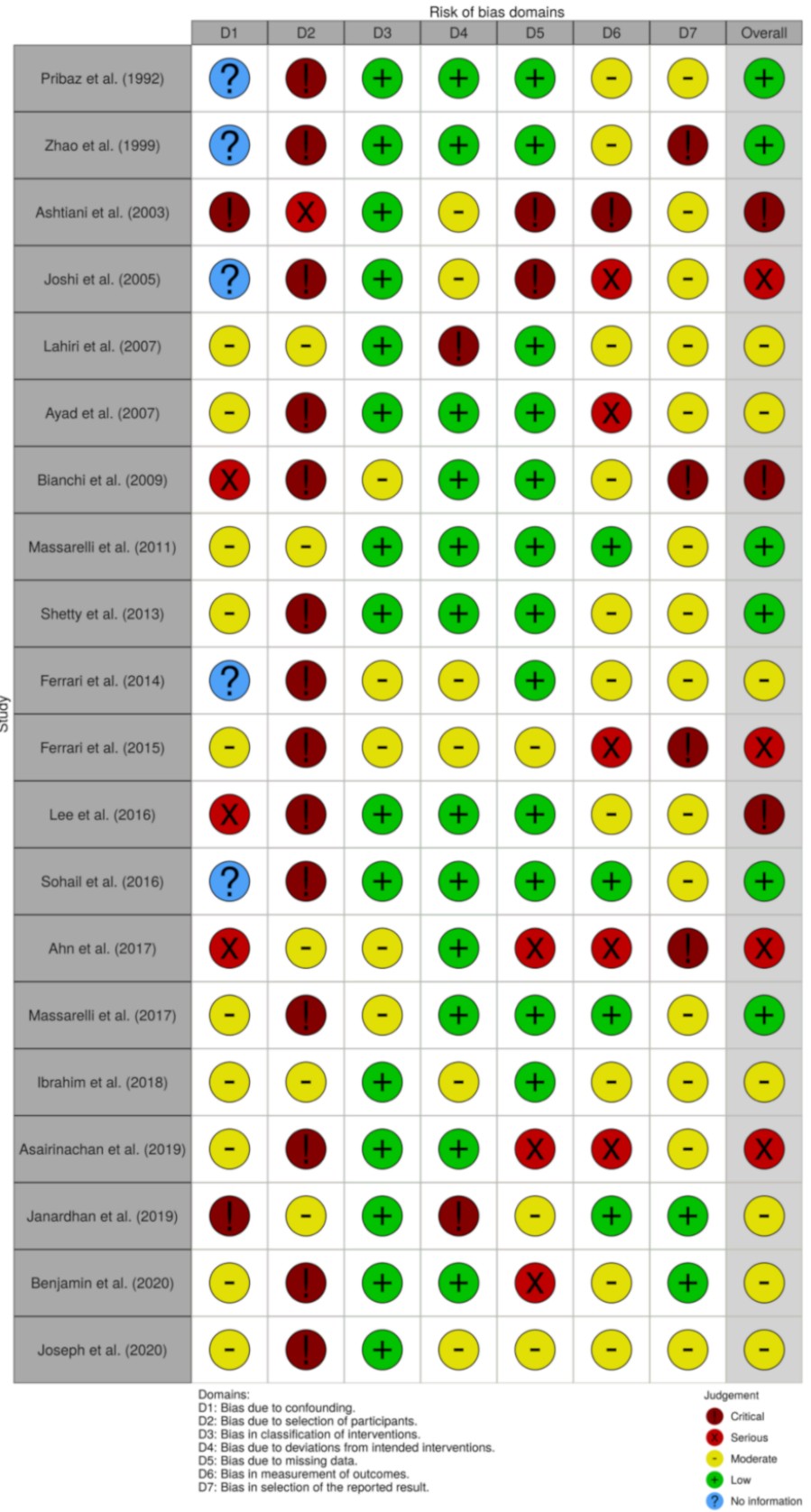

**Figure 2.** ROBINS-I Traffic Light Plot bias assessment.

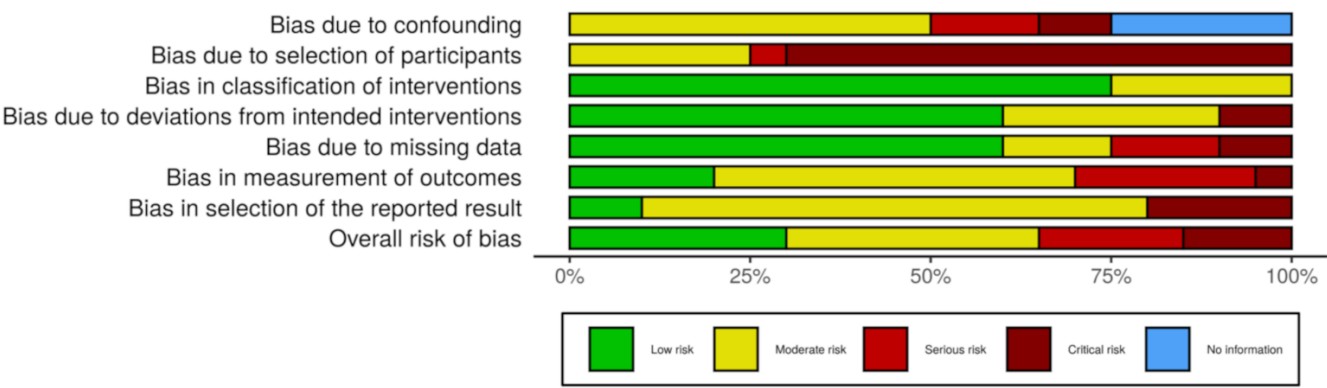

**Figure 3.** ROBINS-I Weighted Summary Plot bias assessment.

**Table 1.** Synthesis of results.

| Author | FAMM | i-FAMM | FAMM Viability | i-FAMM Viability |
|---|---|---|---|---|
| Pribaz et al. (1992) | 18 | | 1 flap loss 1 marginal necrosis | |
| Zhao Z et al. (1999) | | 12 | | None |
| Ashtiani et al. (2005) | 22 | | 2 partial necrosis 1 total necrosis | |
| Joshi A. et al. (2005) | 2 | 15 | | 1 marginal necrosis |
| Lahiri A. et al. (2007) | 16 | | 2 flap loss 2 marginal necrosis | |
| Ayad et al. (2008) | 61 | | 15 partial necrosis | |
| Bianchi B et al. (2009) | 27 | 9 | 1 total necrosis 1 partial necrosis | |
| Massarelli et al. (2012) | 20 | 30 | | 1 total necrosis |
| Shetty et al. (2013) | 11 | | 2 marginal necrosis | |
| Ferrari et al. (2015) | 12 | | 1 partial necrosis | |
| Ferrari et al. (2015) | 36 | 14 | None | None |
| Lee et al. (2016) | 17 with personal modification (S) | | 1 distal necrosis | |
| Sohail et al. (2016) | 16 | | 1 marginal necrosis | |
| Ahn et al. (2017) | 3 | 3 | 2 partial necrosis | 1 flap loss |
| Massarelli et al. (2017) | | 17 | | None |
| Ibrahim B. et al. (2018) | 55 (29 traditional vs. 26 modified) | | 1 distal necrosis in traditional group | |
| Asairinachan et al. (2019) | 13 | | None | |
| Janardhan et al. (2020) | | 16 | | None |
| Benjamin et al. (2020) | 21 | | None | |
| Joseph et al. (2020) | | 20 | | None |
| Total | 350 | 136 | 5 total necrosis 29 partial necrosis | 2 total necrosis 1 partial necrosis |

**Table 2.** Flap reconstruction details and additional reported complications.

| Author | Anatomical Region to Reconstruct | Defect Average Size (mm) | Reported Complications |
|---|---|---|---|
| Pribaz et al. (1992) | Soft/hard palate, floor of mouth, upper lip, lower lip, nasal mucosa | Not reported | None |
| Zhao Z et al. (1999) | Palate, alveolus, nasal septum, orbit | Not reported | None |
| Ashtiani et al. (2005) | Palate | $12 \times 12 - 40 \times 30$ | None |
| Joshi A. et al. (2005) | Alveolus, lip, floor of mouth, palate | $80 \times 30$ (MAX) | Cheek edema and tightness |
| Lahiri A. et al. (2007) | Hard palate | $32 \times 16$ (MAX) | Palate fistula |

**Table 2.** *Cont.*

| Author | Anatomical Region to Reconstruct | Defect Average Size (mm) | Reported Complications |
|---|---|---|---|
| Ayad et al. (2008) | Floor of mouth | Not reported | Soft tissue infection, submental abscess, cheek abscess, cheek hematoma, tongue tethering by scar formation, trismus caused by inner cheek scar |
| Bianchi B et al. (2009) | Tongue, floor of mouth, palate, cheek mucosa, lip | Not reported | Partial mouth opening limitation, partial compromise of swallowing and speech |
| Massarelli et al. (2012) | Lower lip, upper lip, hard palate, soft palate, tongue, maxillary alveolar ridge, mandibular alveolar ridge, retromolar trigone, pharynx, floor of mouth, uvula, tuber maxillae, tonsillar fossa | $30 \times 20 - 100 \times 70$ | Salivary fistula, temporary marginalis mandibulae nerve palsy, cheek scar contracture |
| Shetty et al. (2013) | Soft palate, hard palate | $10 \times 10 - 20 \times 20$ | Suture dehiscence |
| Ferrari et al. (2015) | Hard palate, superior alveolar crest, upper lip, lip commissure, nasal septum, lateral nasal wall, inferior conjunctiva | Not reported | Nasal obstruction, palatal wound dehiscence, donor site dehiscence, lagophthalmos |
| Ferrari et al. (2015) | Tongue, floor of mouth | Not reported | Would dehiscence, neck infection due to floor of mouth dehiscence |
| Lee et al. (2016) | Palate | $10 \times 10 - 33 \times 33$ | None |
| Sohail et al. (2016) | Palate | Not reported | Infection, dehiscence, eating problems, speaking difficulty, donor site scar |
| Ahn et al. (2017) | Tongue, floor of mouth, upper alveolar ridge, lower lip | $53 \times 38$ | Minimal limitation of mouth opening, transient marginal mandibular nerve palsy |
| Massarelli et al. (2017) | Soft palate, hard palate, retromolar trigone, hemipharynx, uvula | $40 \times 30 - 70 \times 60$ | Minor suture dehiscence |
| Ibrahim B. et al. (2018) | Not reported | Not reported | Cheek hematoma |
| Asairinachan et al. (2019) | Base of tongue, tonsil, posterior pharyngeal wall | Not reported | Donor site bleeding, donor site infection, severe dysphagia, mild to severe dysarthria |
| Janardhan et al. (2020) | Tongue, palate, floor of mouth | Not reported | Minor suture dehiscence |
| Benjamin et al. (2020) | Tongue | Not reported | Hemorrhage, wound infection, fistula, oral abscess, mild speech and swallowing impairment |
| Joseph et al. (2020) | Lateral tongue | $60 \times 40$ | Neck hematoma, neck infection, mandibular nerve paresis, mouth opening limitation, dysarthria, dysphagia |

## 4. Discussion

### 4.1. Summary of Evidence

From the myomucosal side of the cheek, several flaps are available to reconstruct different small to medium sized defects of the oral cavity such as the floor of the mouth, tongue, hard and soft palate. Bozola was the first one identifying the myomucosal cheek as a donor site for intraoral reconstruction [4]. The author described the axial buccinator musculomucosal flap, pedicled on the buccal artery, with a posterior mucosal pivot at the maxillary tuberosity and a horizontal axis on the oral commissure. In 1991, Carstens et al.,

in contrast with Bozola's beliefs, reported that the buccal artery was not the main blood supply of the buccinator, but the facial artery seemed to be preeminent instead [28]. On this basis, Carstens proposed the anteriorly based buccinator myomucosal island flap based on the facial artery and vein [29,30]. In 1992, Pribaz et al. proposed the facial artery musculomucosal flap, an axial myomucosal flap centered on the course of the facial artery with an orthograde flow (inferiorly based) or a reverse flow (superiorly based). In 1999, Zhao et al. described two buccinator myomucosal island flaps, with two different vascular patterns: the buccinator myomucosal neurovascular flap posteriorly based supplied by the buccal artery and the buccinator myomucosal reversed-flow arterial island flap superiorly based supplied by the lateral nasal artery, a terminal branch of the facial artery, with a reverse flow supply [6].

The main difference between FAMM and i-FAMM flaps is represented by the need, for the former, of a temporary post-operative bite block and the need for secondary pedicle section surgery, at least in edentulous patients. i-FAMM, on the contrary, has the advantage of avoiding the aforementioned secondary surgery, although the harvesting is technically challenging compared to FAMM.

The present systematic review was performed in order to evaluate flap viability between FAMM and i-FAMM considering flap loss and flap marginal necrosis. During the article evaluation process, the reviewers encountered a problem regarding flap nomenclature. This issue was already raised by a review performed by Massarelli et al. who tried to suggest a rational and simplified nomenclature for the buccinator myomucosal flaps. Massarelli et al. identified the flap names based on the vascular pedicle: for example, the reverse FAMM flap described by Pribaz, with the axial pattern based on the facial artery and a reverse flow, was identified as the nasal artery myomucosal (NAMM) flap because the reverse flow was based on the nasal artery.

In the review performed by Massarelli et al. the most confused nomenclature was the one related to the BAMM flap. First described by Bozola et al., it is an axial flap based and centered on the buccal artery, an internal maxillary artery branch, extending from the maxillary tuberosity to the oral commissure. It can be used to close small mucosal defects of the posterior hard palate, soft palate and maxillary alveolus. The nomenclature of this flap is confused and imprecise. Vague terms such as Bozola's flap or buccal flap or the improper FAMM flap have been often used. The term buccal artery myomucosal flap is precise and unequivocal. Moreover, the acronym BAMM is similar to FAMM, and it could be successfully accepted and avoid confusion resulting from the use of incorrect or confused terms. For the aforementioned reasons, in order to perform an objective evaluation, the reviewers carefully examined the description of the harvesting technique of all selected papers. The flaps where the pedicle was not identified and where the flap was almost with random vascularization rather than axial were excluded. All BAMM flaps with the identification of the buccal pedicle were considered as FAMM.

However, during the review process, we also faced some personal modifications of the FAMM flap, such as those proposed by Lee or Ibrahim; in both cases the flaps were based on the facial artery [19,23]. Lee's modification was based on creating space in the retromolar trigone and palate to accommodate an inferiorly based pedicle in order to avoid secondary division. The modification proposed by Ibrahim had the aim of avoiding secondary surgery for pedicle interruption and flap inset. The procedure forecasts the anterior incision of the flap to be extended into the alveolar crest in order to reach the defect; the base of the flap is then dissected in a subperiosteal plane over the alveolar crest all the way to the floor of mouth. Additional mucosa can be resected posterior to the defect if needed to accommodate any redundancy. The flap, now in continuity with the defect, can be rotated in place with no intervening bridge; the gingiva is incorporated to the base of the flap. Both modifications, performed by Lee and Ibrahim, in the present review were incorporated in the FAMM flap group.

The absolute number of complication cases analyzed is low; nevertheless, the low necrosis rate reported shows how both flaps have an average good survivability. The

higher partial necrosis rate of FAMM flaps could be explained by the harvesting technique: the flap is sometimes grossly harvested, solely considering the facial artery axis. Therefore, marginal tissue could keep random vascularization, and arterial blood flow may not withstand the metabolic demand of the entire area. On the other hand, in cases of island FAMM flap harvesting it is mandatory to perform a meticulous dissection of the facial artery pedicle; a careful dissection of the pedicle could explain the higher survival rate of i-FAMM vs. FAMM.

### 4.2. Limitations

Some limitations can be identified in this review. The wide and confused nomenclature that arose from literature analysis directly influenced the viability assessment. The so-called BAMM flap is sometimes intended as an axial flap based on buccal artery performed with the pedicle identification, while some other authors harvest it without pedicle identification, using only theoretical pedicle orientation. We carefully evaluated all the flap harvesting techniques published and excluded those with no pedicle identification; therefore, in the total count of FAMM flaps there were also some flaps identified as BAMM by the authors but with the identification of the vascular pedicle. Moreover, the realistic viability rate of this flap may be highly biased due to the intrinsic nature of surgical case reports in literature, mostly affected by positive-outcome bias. The high scores encountered during bias evaluation of patient selection, outcome measurement and selection of reported results sections uphold our concerns.

### 5. Conclusions

The iFAMM flap retains analogous total necrosis rate and a lower partial necrosis rate compared to FAMM flap. Although the technique is more difficult to execute, the advantages represented by lower complication rate, higher flexibility in defect reconstruction and absence of compulsory second surgery could widen surgeons' choices in oral defect reconstructions.

**Author Contributions:** Conceptualization, R.R.; Data curation N.Z.; Formal analysis, R.F. and G.C.; Methodology, E.P.; Project administration G.F.N.; Writing—original draft, R.R; Writing—review and editing, G.L.G. All authors have read and agreed to the published version of the manuscript.

**Funding:** This research received no external funding.

**Informed Consent Statement:** Not applicable.

**Acknowledgments:** All co-authors have read and approved the final version of the manuscript.

**Conflicts of Interest:** The authors declare no conflict of interest.

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
