# Peer review of "Facial Artery Myomucosal Flap vs. Islanded Facial Artery Myomucosal Flap Viability: A Systematic Review"

_applsci, doi:10.3390/app11094202_

Round 1

Reviewer 1 Report

Well-chosen purpose of the work. Collected extensive material for comparison. Review gives readers good insight into the effectiveness of facial artery-supplied flap reconstruction for filling intraoral defects. The results are described in great detail. A schematic representation of both compared flaps would be good to include as a figure in this study.

Author Response

Thank you for your comment, we hope you are happy with the reviews. In the Review there are already many tables and figures so we didn't want to make the paper too heavy.

Reviewer 2 Report

This is a good and interesting paper,however I propose some minor revision before publication:

1,I do not recommend any abbreviation in the titel.

2,In Table 1.we find all the necessary data. It is therefore not required to go into detail of the individual studies in the Results chapter. Table 1 is sufficient.

3, On the other hand, an additional table may be necessary detailing the size of the defects, the anatomical regions (e.g. the palate, floor of the mouth etc.), and the complications.

4, The text that appears in rows 338-341 would be better placed in the Results chapter.

Following the revision mentioned above, I recommend the manuscript for publication.

Author Response

Dear Reviewer thank you for your comment. We have followed all your suggestions and we hope they are good for you.